# OpenReview forum: "FiRE: Fine-grained Ranking Evaluation for Machine Translation"
_ICML.cc/2026/Conference — ICML 2026 regular_

### Official Review · Reviewer_PRNM · 2026-02-27

**Soundness:** 2
**Presentation:** 3
**Significance:** 2
**Originality:** 2
**Overall Recommendation:** 4
**Confidence:** 4

**Summary:**

This paper proposes a prompt-engineered, fine-grained machine translation ranking method, along with a human-annotated, reference-free benchmark for fine-grained ranking evaluation.

**Compliance With Llm Reviewing Policy:**

Affirmed.

**Final Justification:**

I would give a weak accept because I believe this work introduces a new MT evaluation framework, which is much needed in the current MT field, where reliable evaluation is lacking and hindering MT development. However, I recommend reconsidering the novelty of the method when making the final decision on acceptance or rejection. While I have raised my score to 4, I still find the method to be somewhat trivial.

**Key Questions For Authors:**

* Why did you choose the three criterion dimensions—Faithfulness, Fluency, and Consistency of Style? Given that there are many scoring dimensions in MQM, how did you select these three?
* I’d like to ask if you use the overall ranking prompt as the final ranking and use the fine-grained prompts solely for interpretability, or do you rely on the majority vote from the fine-grained prompts? Also, is there a possibility of conflict where the majority of fine-grained prompts rank T1, but the overall ranking prompt selects T2?

**Limitations:**

See weaknesses and questions

**Strengths And Weaknesses:**

### Strengths

* A good attempt to incorporate fine-grained methods into the MT evaluation ranking paradigm.
* Proposes a human-annotated MT evaluation benchmark.

### Weaknesses

* The method feels rather trivial: it mainly incorporates several fine-grained, criterion-based ranking prompts into off-the-shelf LLMs, without substantial methodological novelty beyond prompt design.
* The so-called interpretable, fine-grained feedback is bounded by the chosen prompting dimensions, and it is not particularly contributive to the current MT evaluation field.
* The comparison is not fair: you use DeepSeek as the base model, but do the baselines M-MAD and GEMBA-MQM use the same DeepSeek base model? As far as I know, these two methods are not limited to DeepSeek, and this experimental setup is not very convincing.

---

> ### Author Rebuttal · Authors · 2026-03-31
>
> ## Response to Reviewer PRNM
>
> We thank the reviewer for the thoughtful feedback.
>
> **W1 (method seems trivial).** We agree that the methodological novelty of FiRE is not in training a new evaluator, and we will revise the paper to make this clearer. Our contribution is instead the problem formulation and benchmark: FiRE defines criterion-conditioned, reference-free pairwise MT evaluation, and the benchmark directly annotates this setting with explicit criteria and ties allowed. Importantly, the fine-grained signals are not only descriptive: the synthesized FiRE decision also improves over the direct ranking prompt (65.3 vs. 64.3 on EN→ZH; 70.1 vs. 66.7 on RU→ZH), showing that criterion decomposition is useful for overall ranking as well.
>
> **W2 (fine-grained feedback is bounded ).** We respectfully disagree that the fine-grained feedback is of limited value. Its purpose is not merely to provide extra explanation text, but to enable actionable diagnosis. In Section 4.6, FiRE reveals system differences that a single holistic score obscures, such as systems that are relatively fluent but weaker on faithfulness, or systems whose strengths differ across EN→ZH and RU→ZH. We will revise the paper to emphasize this diagnostic value more clearly.
>
> **W3 (different LLM backbones).** Thank you for raising this important point. In the main experiments, we followed the original implementations of the baselines: GEMBA-MQM with gpt-3.5-turbo and M-MAD with gpt-4o-mini. We agree that this was not explicit enough in the submission. To address fairness directly, we re-ran both baselines on our benchmark with a shared backbone, gpt-4o-2024-11-20. Interestingly, performance does not improve and generally decreases. We believe this is consistent with the design of the original methods. Both GEMBA-MQM and M-MAD are prompt-and-pipeline coupled methods rather than backbone-invariant algorithms: their performance depends not only on the backbone, but also on fixed few-shot prompting, debate/parsing procedures, and score aggregation. Replacing the backbone without re-tuning these components can therefore hurt calibration on pairwise criterion benchmark. We also made exploratory runs with R1/QwQ, but GEMBA-MQM’s post-processing is not robust to thinking-style outputs. This is also consistent with their substantial cross-language variance in MQM experiments, where they perform relatively better on high-resource EN→DE but much worse on JA→ZH and HE→EN. Overall, these additional results suggest that the gap between FiRE and these baselines is unlikely to be an artifact of an unfair backbone choice. Due to space, tables are shown in https://anonymous.4open.science/r/Rebuttal-18B7/Tables.png.
>
> **Q1 (criterion selection).** We did not choose these three as an arbitrary subset of MQM tags. Rather, we selected three broad, complementary, and practically meaningful dimensions that are well motivated in prior MT evaluation and user-preference research: semantic fidelity/adequacy, fluency/naturalness, and stylistic preference [1,2,3]. Recent multilingual preference-alignment work makes a similar point: translation quality is not only about accuracy, but also about aligning with human preferences in expression and style [3]. By contrast, MQM is highly valuable for fine-grained error analysis, but its taxonomy is intentionally granular and designed to support task-specific issue selection rather than direct use of all tags at once [4]. We therefore use three higher-level criteria as a compact and practical interface for direct reference-free pairwise annotation, while still preserving compatibility with MQM through criterion-level tag mapping for external validation.
>
> **Q2 (overall vs. fine-grained prompts).** These are two separate variants in the paper. As specified in Sec. 4.5, DeepSeek-R1-Direct-Rank elicits a single overall judgment using one holistic ranking prompt, while DeepSeek-R1-FiRE synthesizes the three fine-grained judgments into an overall decision. FiRE does not combine an overall prompt with the fine-grained prompts. Specifically, it first counts how many criteria favor A vs. B, then resolves ties in the order faithfulness > fluency > consistency of style, and outputs E only when all three criteria are ties. Therefore, the direct overall prompt and the synthesized FiRE decision disagree; this is why we report them separately. Empirically, synthesized FiRE decision aligns better with human judgments than the direct overall prompt. We will revise to make this distinction and the synthesis procedure more explicit.
>
> [1] Evaluating user preferences in machine translation using conjoint analysis. (Kirchhoff, K. et al.) [2] Personalized machine translation: Predicting translational preferences. (Mirkin, S. et al.) [3] PMMT: Preference Alignment in Multilingual Machine Translation via LLM Distillation. (Sun, S. et al.) [4] Multidimensional quality metrics: a flexible system for assessing translation quality. (Lommel, A. R. et al.)

---

> > ### Author Rebuttal · Reviewer_PRNM · 2026-04-01
> >
> > Thank you for the authors’ detailed rebuttal.
> >
> > For W1, as I noted in the strengths, I acknowledge the contributions in problem formulation and benchmark construction. However, I am still not convinced that the method itself is sufficiently distinct from prior work. There has already been substantial prompt-engineering work on fine-grained MT evaluation, and the claim that this is the first to introduce such an approach into pair-wise MT evaluation does not, in my view, establish strong methodological novelty. In addition, pair-wise ranking is already a common setup in reward modeling, so the novelty seems to lie mainly in the formulation rather than the method.
> >
> > For W2 and Q1, FiRE is currently defined over only three dimensions. I remain concerned about its extensibility. What happens if new dimensions are introduced? Would the overall evaluation pipeline need to change? Why were only these three dimensions selected, and why not include more? Even if MQM is fine-grained, it still includes higher-level dimensions. This raises the question of whether the current design is inherently limited in scalability and generalization.

---

> > > ### Author Response · Authors · 2026-04-07
> > >
> > > We sincerely thank the reviewer for the thoughtful follow-up questions, which helped us sharpen both the scope and the contribution claim of the paper.
> > >
> > > We agree that FiRE should not be framed as a new trained evaluator, nor as a novelty claim about pairwise ranking itself. Our intended contribution is narrower but, we believe, still meaningful for MT evaluation. FiRE should be understood as a **fine-grained pairwise evaluation framework for MT**, rather than merely a prompt-engineered variant of pairwise ranking. More specifically, compared with direct pairwise ranking, FiRE offers three practical benefits: richer criterion-level diagnostic information, a more reliable aggregated overall ranking, and better alignment with human judgments. Concretely, this contribution comes from combining **fine-grained, reference-free pairwise evaluation**, **an explicit aggregation mechanism for overall ranking**, and a **benchmark directly annotated for pairwise evaluation**, with criterion-level labels and ties.
> > >
> > > To directly address your concern about extensibility, we extended FiRE with a fourth criterion, **locale convention**, which captures whether a translation follows target-locale conventions in form and presentation, such as punctuation and quotation styles, date and number formatting, units, and other conventionalized target-language expressions. We denote this variant as **FiREplus**. Due to space, tables are shown in *https://anonymous.4open.science/r/FR-5C5D/Tables.md*. Importantly, this extension does not require redesigning the overall evaluation pipeline: the criterion-level evaluation stage remains unchanged, and the overall decision is obtained by incorporating the additional criterion into the same aggregation procedure. We newly annotated 250 EN→ZH pairs with three professional annotators. On this new dimension, the human annotation achieves κ = 55.3, and as shown in `Table 1` , DeepSeek-R1-FiREplus reaches 77.6% agreement with human annotations (QwQ-FiREplus: 60.0%). On this newly annotated 250-pair subset, adding the new criterion not only expands the evaluative scope without requiring any redesign of the pipeline, but also improves the aggregated overall ranking for both evaluated backbones. More importantly, as shown in `Table 2`, after introducing the new dimension, the aggregated overall ranking remains competitive with the original FiRE and still consistently outperforms Direct-Rank on the full ranked data in EN→ZH and RU→ZH. **This suggests that the framework is not hard-coded to exactly three dimensions, and that adding a new criterion does not require changing its basic evaluation paradigm.**
> > >
> > > Regarding why we started from faithfulness, fluency, and consistency of style, we chose them as a compact set of broad, complementary, and practically meaningful dimensions, rather than as a complete taxonomy. **As stated in Sec. 3.1, these criteria were selected to capture different high-level aspects of translation quality in a way that is suitable for reference-free pairwise ranking.** In particular, **faithfulness** reflects whether the translation accurately preserves the source meaning, including whether the model introduces hallucination, omission, or distortion; **fluency** reflects target-side generation quality, including naturalness, readability, and the extent to which the output avoids translationese; and **consistency of style** reflects whether source-side stylistic properties are appropriately preserved across languages, which relates to the model’s cross-lingual style transfer ability. Taken together, they provide a richer and more practically meaningful assessment than a single holistic judgment, while remaining compact enough for direct pairwise annotation and evaluation. We agree that MQM is more comprehensive and highly valuable for fine-grained error analysis, but its role is different from ours. FiRE is intended as a **compact fine-grained ranking framework for pairwise evaluation**, rather than a full hierarchical error taxonomy. In this sense, the FiREplus result is also informative, because it shows that the initial three dimensions are a starting point rather than a hard limit.
> > >
> > > In sum, these additional results and clarifications help us state the contribution more precisely. In the revision, we will make clearer that FiRE’s contribution lies in a **benchmark** directly annotated for pairwise MT evaluation, **evidence** that fine-grained decomposition can support stronger overall ranking than a single direct prompt, and new FiREplus results showing that the framework is **extensible** without redesigning the overall pipeline. We will also clarify that the initial criteria were chosen to instantiate a compact yet extensible framework for fine-grained pairwise MT evaluation, rather than to define a fixed or exhaustive taxonomy of translation quality. We hope these clarifications help resolve the reviewer’s remaining concerns in the final assessment.

---

### Official Review · Reviewer_Gxta · 2026-03-07

**Soundness:** 4
**Presentation:** 4
**Significance:** 4
**Originality:** 3
**Overall Recommendation:** 5
**Confidence:** 4

**Summary:**

The paper proposes a novel preference-based way to measure the quality of machine translation. The proposed metric, called FiRE, takes as input a source sentence and a pair of translations (possibly produced by different MT models) and uses an LLM to determine which of the two translations are better. The authors demonstrate that the agreement between FiRE and human quality preferences is better than for competing metrics. FiRE assesses quality along three dimensions: fidelity, fluency and consistency of style, which provides better insight into translation quality. The authors also contribute the first human-annotated, reference-free benchmark specifically for fine-grained pairwise ranking, covering English-to-Chinese and Russian-to-Chinese directions.

**Compliance With Llm Reviewing Policy:**

Affirmed.

**Final Justification:**

Thanks to the authors for the clarifications. I keep my (already high) score.

**Key Questions For Authors:**

I couldn't find any indication of whether the authors are planning to release the benchmark.

Also a few minor comments:

- The abstract introduces MQM. I believe it should be spelled out (if space permits).
- L090: Did you mean "test" instead of "testify"?
- L091: You state that you "use DeepSeek-R1 as FiRE backbone to evaluate...". Is this because this model showed the best agreement with humans? Either way, I think it would be better to state this right there, to help the reader.
- L145-146 "we also incorporate an overall quality criterion". Is this a simple average of the thee criteria?

**Limitations:**

For high-resource languages, the paper only considers two pairs: `en-zh` and `ru-zh`.

**Strengths And Weaknesses:**

# Soundness

The paper is technically sound, the claims are adequately supported by the extensive experiments and evaluations.

**Strengths**:

- Unlike "black-box" regression scores (e.g., COMET), FiRE provides a rationale for its rankings. This allows developers to see, for instance, if a model like ALMA-13B-R is fluent but prone to hallucinations (low faithfulness).
- The authors demonstrate that FiRE outperforms state-of-the-art metrics like MetricX-24-XXL in aligning with human preferences across several datasets.
- The study is in-depth, testing seven different LLMs (open- and closed-source) and exploring the impact of "easy" vs. "hard" data points.
- The authors create a human-annotated dataset with 12,800 annotations and substantial inter-annotator agreement, which is a significant contribution to the MT community

**Weaknesses**

- While not discussed, using an LLM (especially a reasoning model like DeepSeek-R1) for pairwise comparison is significantly more computationally expensive and slower than traditional metrics like BERTScore or COMET. This may limit its use for real-time evaluation or massive datasets.
- The paper honestly reports significant position bias (where the order of translations changes the judgment) and self-enhancement bias (where models favor their own outputs).

# Presentation

The paper is written clearly and easy to follow. I especially appreciate the concise and well-structured Related Work section, helping the reader understand the problem that the proposed method aims to solve. The results are well-presented, providing extensive additional detail in the appendices.

# Significance

Machine translation research has been dominated by overlap-based (e.g. BLEU) and regression-based (e.g. BLEURT) quality metrics. While certainly useful and easy to evaluate, they only provide a rather crude measure of translation quality, and there is a need for new metrics that correlate more reliably with human preferences. FiRE addresses this important gap.

# Originality

While using an LLM as a judge is not novel per se, three isn't much (if any) published research into how LLMs could be used for fine-grained MT quality evaluation.

---

> ### Author Rebuttal · Authors · 2026-03-31
>
> ## Response to Reviewer Gxta
>
> Thank you for the positive assessment and for recognizing the value of FiRE’s fine-grained, reference-free pairwise evaluation. We are especially encouraged that the reviewer found the benchmark and the empirical study significant.
>
> **W1 (computational cost).** We agree that LLM-based pairwise evaluation is more computationally expensive than lightweight learned metrics such as COMET-style scorers, and we will make this limitation more explicit in the revision. Our goal, however, is not to replace low-cost metrics in latency-sensitive settings, but to provide a higher-fidelity and more interpretable evaluator for meta-evaluation and system diagnosis. FiRE is also backbone-agnostic rather than tied to a single expensive model: our ablation shows that multiple LLM backbones are competitive, while reasoning-oriented judges are generally more robust. We use DeepSeek-R1 in the main results because it gives the strongest and most stable agreement with human judgments in our ablation.
>
> **W2 (position bias and self-enhancement bias).** We agree that these are important limitations. Importantly, our paper explicitly studies both position bias and self-enhancement bias, rather than treating the evaluator as a black box. We show that position bias is substantial, and that GPT-4o and Qwen-series evaluators also exhibit clear self-enhancement tendencies. In the rebuttal, we further conducted an additional easy/hard analysis of position bias (please refer to our response to Reviewer fAuL), and in the revision we will make the mitigation discussion more explicit, e.g., using reasoning-based models, and avoiding same-family evaluators when assessing their own outputs.
>
> **Q1 (open source release plan).** To clarify the reproducibility aspect, we plan to release the benchmark, code, and annotation guidelines after the anonymous review period.
>
> **Q2 (minor comments).** We will incorporate the reviewer’s helpful suggestions in the revision: 1) spell out MQM on first use in the abstract; 2) correct “testify”; 3) clarify earlier that DeepSeek-R1 is selected because it performs best and most stably in our ablation; 4) clarify that the overall FiRE decision is not a simple average. As already specified in Sec. 4.5 (L327), it first compares how many criteria favor A vs. B, and in tied cases applies a lexicographic tie-break in the order faithfulness > fluency > consistency of style.
>
> **Limitation (language coverage).** We agree that our original human-annotated benchmark is limited to EN→ZH and RU→ZH. To strengthen the evidence beyond these two directions, we have now added MQM24 EN→DE, another high-resource language pair, in the rebuttal. This complements the cross-dataset evidence already included in the paper on MQM23 HE→EN and MQM24 JA→ZH, and further shows that the method is not restricted to the two benchmark directions.
>
> ### **Results on MQM24 EN→DE**
>
> |  | Faithfulness | Fluency | Cons. of Style | Overall |
> | --- | --- | --- | --- | --- |
> | *Regression-Based* | ——— | ——— | ——— | ——— |
> | KIWI-XL | - | - | - | 65.3 |
> | KIWI-XXL | - | - | - | 66.7 |
> | XCOMET-XL | - | - | - | 64.6 |
> | XCOMET-XXL | - | - | - | **67.9** |
> | MetricX-24-XXL | - | - | - | 67.4 |
> | *Error-Based* | ——— | ——— | ——— | ——— |
> | GEMBA-MQM | 65.3 | 55.5 | 53.4 | 63.5 |
> | M-MAD | 62.6 | 52.5 | 49.5 | 59.7 |
> | *Ranking-Based* | ——— | ——— | ——— | ——— |
> | MT-Ranker-Base | - | - | - | 62.0 |
> | MT-Ranker-Large | - | - | - | 63.9 |
> | MT-Ranker-XXL | - | - | - | 64.5 |
> | QwQ-FiRE | **67.0** | **59.4** | **53.7** | 66.3 |

---

> > ### Author Rebuttal · Reviewer_Gxta · 2026-04-02
> >
> > Thanks to the authors for the clarifications. I keep my (already high) score.

---

> > > ### Author Response · Authors · 2026-04-07
> > >
> > > We sincerely thank the reviewer for the very positive assessment and the helpful suggestions throughout the discussion. We are especially encouraged that the clarifications addressed your concerns.
> > >
> > > In the revision, we will incorporate the points discussed in the rebuttal, including clarifying the motivation for selecting DeepSeek-R1 as the main FiRE backbone, addressing minor presentation issues, and strengthening the discussion of limitations such as computational cost, pairwise evaluator bias, and language coverage. We will also include the additional MQM24 EN→DE evidence to further strengthen the generalization discussion.
> > >
> > > We greatly appreciate your careful reading and supportive feedback!

---

### Official Review · Reviewer_SPgN · 2026-03-10

**Soundness:** 3
**Presentation:** 3
**Significance:** 2
**Originality:** 2
**Overall Recommendation:** 3
**Confidence:** 4

**Summary:**

This paper presents FiRE, a fine-grained ranking framework for reference-free machine translation (MT) evaluation, which conducts multi-criterion pairwise comparison of translation outputs and aggregates dimension-specific judgments into an overall ranking decision.

The authors also build the first human-annotated benchmark tailored for fine-grained reference-free MT ranking evaluation. Experiments show FiRE outperforms mainstream existing MT evaluation metrics in alignment with human preferences, and the paper further analyzes biases in LLM-based evaluators.

**Compliance With Llm Reviewing Policy:**

Affirmed.

**Final Justification:**

I suggest the authors putting more efforts in presenting a solid research. Besides the position bias that may affect the reliability of the proposed approach. I still think the approach itself is not novel enough, while the dataset and analysis might be more useful.

**Key Questions For Authors:**

Q1. What specific large language models (LLMs) were used to implement the GEMBA-MQM and M-MAD baselines in your experiments? The paper only mentions adopting their original implementations but does not disclose the underlying LLM backbones, which is critical for reproducing the baseline results and fairly comparing their performance with FiRE.

Q2. The authors do not disclose key implementation details of MQM benchmark experiments (Section 5.3). For example, how translation pairs were sampled from MQM datasets ?

**Limitations:**

yes

**Strengths And Weaknesses:**

Strengths

S1. The authors conduct thorough comparisons across three dominant MT evaluation paradigms (regression-based, error-based, ranking-based). The experiments cover multi-dimensional performance validation, LLM backbone ablation studies, difficulty-stratified robustness analysis, and targeted bias quantification.

S2. Even though the proposed method is aligned with the design of the authors' in-house benchmark, FiRE achieves state-of-the-art performance on out-of-domain MQM datasets (including low-resource language pairs), demonstrating its robust generalization ability. The criterion-driven pairwise comparison design of FiRE provides a clear and actionable direction for improving error-based evaluation frameworks.

S3. The benchmark built in this work features high-quality human annotations across four criteria, substantial inter-annotator reliability, and balanced coverage of translation quality levels, providing a standardized testbed for future research on fine-grained MT evaluation.


Weaknesses

W1. Limited performance gain of the fine-grained design over direct ranking

The core novelty of this work is the fine-grained, multi-criterion pairwise comparison design to address the lack of interpretability in existing ranking methods. However, experimental results show that FiRE delivers only marginal improvements in overall ranking agreement over a simple direct ranking baseline. Meanwhile, the pairwise ranking paradigm for MT evaluation and open-ended tasks has been widely explored in prior work [1,2,3,4,5], and is not a novel contribution of this paper. This limited performance gain raises questions about the practical value of the fine-grained design when used solely for overall ranking optimization.

W2. Incomplete MQM benchmark experiments

The MQM experiments, which are critical for verifying FiRE's generalization ability, have notable limitations in completeness and transparency. First, the experiments only cover two language pairs and use a single LLM backbone for FiRE, lacking validation on mainstream high-resource language pairs and other tested LLM backbones, which reduces the comprehensiveness of the generalization analysis.

W3. Ambiguity of the fine-grained evaluation dimensions

The consistency of style dimension suffers from inherent subjective ambiguity, as evidenced by its noticeably lower inter-annotator agreement (Fleiss' kappa) compared to faithfulness and fluency.

W4. Insufficient analysis of position bias

The authors report substantial position inconsistency in LLM evaluators, with judgment changes in over 1/3 of cases when translation order is swapped. However, they do not analyze how this inconsistency affects the stability of FiRE's performance, nor do they provide a rigorous explanation for why FiRE still outperforms all baselines despite this issue.

W5. Poor scalability for system-level ranking with large numbers of MT systems

FiRE relies on exhaustive pairwise comparison of all MT system combinations to generate system-level rankings, leading to quadratic growth in inference cost as the number of systems increases. For mainstream large-scale MT benchmarks that often include 10+ participating systems, this design brings prohibitive computational overhead and severely limits the method's scalability and practical applicability in real-world large-scale evaluation scenarios.

[1] MT-Ranker (as cited by the authors in the paper)
[2] Enhancing Human Evaluation in Machine Translation with Comparative Judgement (Song et al., ACL 2025)
[3] Generative Judge for Evaluating Alignment (Li et al., ICLR 2024)
[4] Judgelm: Fine-tuned large language models are scalable judges (Zhu et al., ICLR 2025)
[5] Learning llm-as-a-judge for preference alignment (Ye et al., ICLR 2025)'

---

> ### Author Rebuttal · Authors · 2026-03-31
>
> ## Response to Reviewer SPgN
>
> We thank the reviewer for the careful reading and constructive feedback. We are encouraged that the reviewer recognizes the breadth of our comparisons, the quality of the benchmark, and the strong empirical performance on both our benchmark and MQM datasets. We address the concerns below.
>
> **W1 (limited gain over direct ranking).** We agree that pairwise ranking itself is not the novelty of our work, and we will revise the paper to make this point explicit. Our main contribution is instead a reference-free, criterion-based fine-grained pairwise evaluation framework, together with the first human-annotated benchmark tailored to this setting. We also agree that, if one looks only at a single overall label, the gain over direct ranking is moderate on EN→ZH. However, the main value of FiRE is not merely a stronger direct-ranking prompt: it provides interpretable criterion-level judgments and actionable diagnosis, while still yielding consistent gains in overall agreement. We will revise the contribution statement to emphasize this more precisely.
>
> **W2 (incomplete MQM experiments).** Thank you for this suggestion. We agree that the original MQM section was not comprehensive enough. To strengthen the generalization analysis, we have added experiments on MQM24 EN→DE, a mainstream high-resource language pair, in addition to the original JA→ZH  and HE→EN settings. Due to the length limit, please find detailed EN→DE results in our response to Reviewer Gxta. We note that FiRE already generalized beyond our in-house benchmark to two external MQM datasets, and the new EN→DE result further strengthens this point.
>
> **W3 (ambiguity of consistency of style).** We agree that consistency of style is the most subjective of the three dimensions, and we will clarify this limitation more explicitly. At the same time, the agreement remains substantial even under our more challenging 3-annotator, 3-class setting with ties allowed (κ = 0.57 for EN→ZH and 0.62 for RU→ZH). Moreover, this dimension is not left unconstrained: our protocol operationalizes it through Tone Matching, Emotional Preservation, and Writing Style, which reduces ambiguity and improves annotation consistency. Empirically, FiRE still achieves strong agreement on this dimension and substantially outperforms error-based baselines, suggesting that the signal is harder but still meaningful and usable.
>
> **W4 (position bias).** We agree that position bias is a limitation and a broader challenge for LLM-based pairwise evaluators. Therefore, we explicitly reported it to surface rather than hide this issue. To further analyze its effect, we conducted an additional easy/hard study of position bias (please refer to our response to Reviewer fAuL for details).
>
> **W5 (scalability for system-level ranking).** We agree that exhaustive all-pairs comparison scales quadratically with the number of systems. However, this is not a requirement of the FiRE formulation itself; rather, it is the evaluation protocol we used for complete system-level analysis in the paper. FiRE fundamentally evaluates a pair of translations. The system-level ranking in Section 4.6 is one application of these pairwise judgments, not a prerequisite for the method. In addition, our normalized Copeland aggregation already supports unbalanced comparison graphs, so exhaustive all-pairs comparisons are not required in principle; subsampled, anchor-based, or tournament-style schedules are compatible with our framework. We will clarify this distinction and discuss scalable comparison schedules as an important future direction.
>
> **Q1 (LLM backbones for GEMBA-MQM and M-MAD).** Thank you for pointing this out. In the main experiments, we followed the original implementations of the two baselines: GEMBA-MQM used gpt-3.5-turbo and M-MAD used gpt-4o-mini. We agree that this was insufficiently explicit. To address fairness, we additionally conducted controlled-backbone experiments with a shared backbone; due to  length limit, please refer to our response to Reviewer PRNM’s W3 for the detailed analysis and results. We will add these implementation details explicitly in the revision.
>
> **Q2 (MQM data construction details).** Thank you for highlighting the missing implementation details. Specifically, our dataset is constructed by generating all pairwise combinations of candidate translations for each MQM source sentence and mapping the original MQM scores to three criteria according to Table 13, consistent with our GEMBA-MQM and MMAD evaluations. We exclude cases where two translations are identical and retain only pairs that exhibit a clear preference, and use them to form the ranked dataset in our experiments. We will clarify the MQM data construction process explicitly in the revision.
>
> Due to the rebuttal length limit, some overlapping details are provided in our responses to other reviewers.

---

> > ### Author Rebuttal · Reviewer_SPgN · 2026-04-03
> >
> > It is still strange for me that a system with position consistency only 0.5-0.8 could have high practical values  in its current form. Since its results could be flipped in many cases just because of the change in input order.

---

> > > ### Author Response · Authors · 2026-04-07
> > >
> > > We sincerely thank the reviewer for continuing this discussion. We fully agree that position bias is an important practical concern, and your comment helped us sharpen this part of the paper. In response, we conducted new swap-based analyses to test whether the observed order sensitivity actually undermines the practical value of FiRE.
> > >
> > > For the swap experiment, we keep the source sentence, evaluator (MT-Ranker or FiRE), and input structure unchanged, and only exchange the positions of translation A and translation B in the input. We then rerun the evaluator on the swapped input. For FiRE, this produces a new set of three criterion-level judgments, which are then synthesized into an overall decision using exactly the same rule as in the paper. For MT-Ranker, this progress directly produces a new overall judgment for meta evaluation. Due to space, results are displayed in *https://anonymous.4open.science/r/final-response-04EA/AllTables.md*.
> > >
> > > **Our new results suggest that position bias is best understood as a broader challenge of pairwise MT ranking, rather than a failure specific to FiRE.** As shown in `Table 1`, MT-Ranker-Large and MT-Ranker-XXL reach only 68.5 and 62.7 position consistency overall, while DeepSeek-R1-FiRE and QwQ-FiRE both reach 76.5. This shows that even dedicated trained ranking models remain substantially sensitive to candidate order, and stronger MT-Ranker variants can be less stable than FiRE under swapping. At the same time, we do not want to downplay the limitation: in the paper, we explicitly reported that position consistency is far from perfect, with average values around 65.6 to 67.4 across criteria.
> > >
> > > More importantly, the practical question is not whether every intermediate criterion label is invariant, but whether the **final overall decision** remains aligned with human judgment after FiRE’s intended aggregation. Here the swap experiment shown in `Table 2` is encouraging. MT-Ranker-Large drops from 60.9 to 44.6 agreement after swapping, and MT-Ranker-XXL drops from 61.6 to 41.9. In contrast, DeepSeek-R1-FiRE changes from 65.3 to 65.9, and QwQ-FiRE from 65.3 to 64.6. So while some criterion-level judgments do change, the final FiRE decision is much more stable.
> > >
> > > To further probe this, in `Table 3` we compare several families of aggregation strategies over the predictions from both candidate orders: A. directly aggregating all six criterion judgments; B. first reconciling the two orders at the criterion level and then aggregating; C. first producing one overall decision per order and then taking overall consensus. The key observation is that the fine-grained aggregation strategies remain close to the original FiRE result. For example, “Criterion Consensus → Aggregation” gives 65.9 for DeepSeek-R1-FiRE and 64.6 for QwQ-FiRE, versus 65.3 and 65.3 in the paper. By contrast, the strategy that first collapses each order into a single overall label and only then takes consensus drops to 54.1 and 53.8. **This suggests that the fine-grained structure is not only more interpretable, but also part of why FiRE is comparatively robust: aggregating multiple criterion-specific judgments absorbs part of the order noise instead of amplifying it.**
> > >
> > > This is also consistent with the original paper’s main result that FiRE (synthesized overall across fine-grained pairwise ranking) is stronger than Direct-Rank (a single direct overall ranking), improving from 64.3 to 65.3 on EN→ZH and from 66.7 to 70.1 on RU→ZH. **Our updated takeaway is therefore narrower but stronger: position bias is real, but it is a task-level challenge for pairwise MT evaluation, not evidence that FiRE lacks practical value.** Once we evaluate the stability of the **final aggregated decision**, FiRE remains robust and clearly useful in practice. We will revise the paper to include these new experiments and make this limitation and its mitigation more explicit.
> > >
> > > Taken together, these additional analyses suggest that FiRE should be understood not simply as another pairwise ranking method, but as a fine-grained pairwise evaluation framework whose advantages over direct pairwise ranking lie in three aspects: it provides **richer criterion-level diagnostic information**, yields a **more robust overall decision under position bias**, and achieves **better alignment with human judgments**. In the revision, we will incorporate the reviewer-suggested related work more explicitly and sharpen this contribution statement accordingly.
> > >
> > > We hope this additional evidence and clarification of our contribution resolves the reviewer’s remaining concern, and that the reviewer will take these new results into account in the final assessment.

---

### Official Review · Reviewer_fAuL · 2026-03-20

**Soundness:** 3
**Presentation:** 3
**Significance:** 3
**Originality:** 2
**Overall Recommendation:** 5
**Confidence:** 4

**Summary:**

The paper proposes a fine-grained ranking evaluation method for MT via reference-less pairwise comparison using LLMs. The assessment dimensions are faithfulness, fluency and style consistency. Seven state-of-the art open-source and closed-source LLMs are assessed as backbones. Six MT LLMs are benchmarked, including specialised MT models. The authors created a human ranking benchmark for their method (English-Chinese, Russian-Chinese, ~3K data points). The proposed approach is compared to a range of recent regression-based, error-based and ranking-based approaches. There is also an assessment for the translations stratified by difficulty, and for the low-resource condition (Japanese-Chinese, Hebrew-English). The models’ positional bias is also assessed. DeepSeekR1 seems to perform the best as an evaluator, seems like GPT-4o is ranked as one of the best translators.

**Compliance With Llm Reviewing Policy:**

Affirmed.

**Final Justification:**

My main concern about the practical value of the framework has been resolved: position bias can be assuaged using averaging, human assessments are also prone to it. I hence increase my score.

**Key Questions For Authors:**

Given that position bias is > 30%, how do these unstable examples relate to the easy and difficult cases defined earlier? Are these the cases where humans mostly disagree as well? Comment on the practical applicability of the framework seen this instability.

**Limitations:**

yes

**Strengths And Weaknesses:**

Strengths:
- Well-written and easy-to-follow paper, impressive amount of detail and experimentation
- There is a good related work overview
- A well-founded proposal of a fine-grained pairwise ranking framework for MT with LLMs. It re-visits the established evaluation tradition in the MT community with LLMs
- Results could be impactful for the MT community, the framework is novel
- Model generalises well across setups with difficult translation language pairs

Weaknesses:
Practical value of the framework requires additional clarification (see my questions below)

---

> ### Author Rebuttal · Authors · 2026-03-31
>
> ## Response to Reviewer fAuL
>
> We thank the reviewer for the encouraging assessment and for highlighting this important practical concern. We agree that position bias is highly relevant to the practical use of LLM-based evaluators, and we therefore conducted an additional analysis to relate position bias to the easy/hard split in Section 5.1.
>
> Our main finding is that position instability does not concentrate only on the human-hard subset. If the performance drop on hard cases were mainly caused by position bias, we would expect position consistency to decrease systematically on hard examples. However, the new results do not show such a monotonic trend: for several models and criteria, hard cases are as stable as, or even more stable than, easy cases. By contrast, our original results do show a clear drop in agreement with human annotations on hard subsets (Table 4 on Page 7). Taken together, these findings suggest that the degradation on hard cases is driven more by the difficulty of aligning with human comparative judgments than by candidate-order sensitivity alone.
>
> We therefore view position bias as an important but partly orthogonal evaluator-side calibration issue, rather than evidence that FiRE loses practical value on difficult examples. Importantly, despite this limitation, FiRE still achieves the strongest agreement with human judgments in overall pairwise evaluation while providing criterion-level diagnostic feedback. In practice, this issue can also be mitigated by evaluating both candidate orders and aggregating the judgments. We will revise the paper to include this additional analysis and clarify this point more explicitly, which we believe will further strengthen the discussion of FiRE’s practical applicability and limitations.
>
> ### **Position consistency on EN→ZH**
>
> | Model | Faithfulness (Easy) | Faithfulness (Hard) | Fluency (Easy) | Fluency (Hard) | Cons. of Style (Easy) | Cons. of Style (Hard) |
> | --- | --- | --- | --- | --- | --- | --- |
> | Qwen | 0.617 | 0.746 | 0.651 | 0.511 | 0.613 | 0.500 |
> | Mistral | 0.596 | 0.644 | 0.671 | 0.596 | 0.744 | 0.579 |
> | GPT-4o | 0.567 | 0.627 | 0.783 | 0.511 | 0.487 | 0.632 |
> | Claude | 0.631 | 0.644 | 0.691 | 0.617 | 0.713 | **0.658** |
> | Gemini | **0.752** | 0.644 | 0.566 | 0.489 | 0.581 | 0.632 |
> | DeepSeek-R1 | 0.745 | 0.712 | **0.816** | **0.809** | 0.719 | **0.658** |
> | QwQ | 0.603 | **0.780** | 0.730 | 0.617 | **0.800** | 0.632 |
>
> ### **Position consistency on RU→ZH**
>
> | Model | Faithfulness (Easy) | Faithfulness (Hard) | Fluency (Easy) | Fluency (Hard) | Cons. of Style (Easy) | Cons. of Style (Hard) |
> | --- | --- | --- | --- | --- | --- | --- |
> | Qwen | 0.687 | 0.541 | 0.546 | 0.622 | 0.506 | 0.500 |
> | Mistral | 0.571 | 0.622 | 0.607 | 0.757 | 0.717 | 0.656 |
> | GPT-4o | 0.699 | 0.730 | **0.791** | 0.838 | 0.639 | 0.562 |
> | Claude | 0.589 | 0.486 | 0.693 | 0.595 | 0.669 | 0.688 |
> | Gemini | **0.773** | 0.730 | 0.589 | 0.730 | 0.614 | 0.656 |
> | DeepSeek-R1 | 0.748 | **0.757** | 0.773 | 0.730 | 0.723 | 0.781 |
> | QwQ | 0.736 | **0.757** | 0.761 | **0.892** | **0.747** | **0.812** |

---

> > ### Author Rebuttal · Reviewer_fAuL · 2026-04-03
> >
> > The authors have fully addressed my concerns regarding the effect of the position bias on the usability of the framework, as well as its relation to example difficulty level.

---

> > > ### Author Response · Authors · 2026-04-07
> > >
> > > We sincerely thank the reviewer for the positive reassessment. We are very encouraged that our additional analysis resolved your concern regarding the practical effect of position bias and its relation to example difficulty.
> > >
> > > In the revision, we will incorporate this easy/hard position-bias analysis and clarify one further practical takeaway from our additional swap-based study. Specifically, the new results suggest that position bias is better understood as a broader challenge for pairwise MT evaluators, rather than a limitation specific to FiRE. Importantly, while some intermediate criterion judgments may change under candidate-order swapping, FiRE’s fine-grained aggregation helps make the final overall ranking more robust. This is also consistent with the paper’s main finding that FiRE yields stronger overall ranking than Direct-Rank.
> > >
> > > We will therefore strengthen the discussion of both this limitation and its practical mitigation in the revision. We sincerely appreciate your careful reading and constructive feedback, which helped us improve the paper!

---

### Decision · Program_Chairs · 2026-04-30

**Decision:**

Accept (regular)

**Comment:**

The paper proposes a fine-grained ranking evaluation method for MT via reference-less pairwise comparison using LLMs. The assessment dimensions are faithfulness, fluency and style consistency. The authors created a human ranking benchmark for their method (English-Chinese, Russian-Chinese, ~3K data points). The proposed approach is compared to a range of recent regression-based, error-based and ranking-based approaches.

All reviewers are generally positive about this work since it outperforms state-of-the-art metrics like MetricX-24-XXL when correlated with human judgement and provide more fine-grained explaination. As Reviewer 6xta pointed out,  "Unlike "black-box" regression scores (e.g., COMET), FiRE provides a rationale for its rankings. This allows developers to see, for instance, if a model like ALMA-13B-R is fluent but prone to hallucinations (low faithfulness)." Minor issues raised by the reviewers still needs to be addressed.